# Evaluation of freely available data profiling tools for health data research application: a functional evaluation review

Ben Gordon [1], Clara Fennessy,[1] Susheel Varma [1], Jake Barrett [1],[1] Enez McCondochie,[2] Trevor Heritage,[2] Oenone Duroe,[2] Richard Jeffery,[2] Vishnu Rajamani,[2] Kieran Earlam,[3] Victor Banda,[4] Neil Sebire[1]

¹Central Team, Health Data Research UK, London, UK
²Inspirata Ltd, Tampa, Florida, USA
³Cystic Fibrosis Trust, London, UK
⁴Neonatal Data Analysis Unit, Imperial College London Neonatal Medicine Research Group, London, UK

**Correspondence to**
Dr Neil Sebire;
neil.sebire@hdruk.ac.uk

## ABSTRACT

**Objectives** To objectively evaluate freely available data profiling software tools using healthcare data.

**Design** Data profiling tools were evaluated for their capabilities using publicly available information and data sheets. From initial assessment, several underwent further detailed evaluation for application on healthcare data using a synthetic dataset of 1000 patients and associated data using a common health data model, and tools scored based on their functionality with this dataset.

**Setting** Improving the quality of healthcare data for research use is a priority. Profiling tools can assist by evaluating datasets across a range of quality dimensions. Several freely available software packages with profiling capabilities are available but healthcare organisations often have limited data engineering capability and expertise.

**Participants** 28 profiling tools, 8 undergoing evaluation on synthetic dataset of 1000 patients.

**Results** Of 28 potential profiling tools initially identified, 8 showed high potential for applicability with healthcare datasets based on available documentation, of which two performed consistently well for these purposes across multiple tasks including determination of completeness, consistency, uniqueness, validity, accuracy and provision of distribution metrics.

**Conclusions** Numerous freely available profiling tools are serviceable for potential use with health datasets, of which at least two demonstrated high performance across a range of technical data quality dimensions based on testing with synthetic health dataset and common data model. The appropriate tool choice depends on factors including underlying organisational infrastructure, level of data engineering and coding expertise, but there are freely available tools helping profile health datasets for research use and inform curation activity.

## INTRODUCTION

Health Data Research UK's (HDR UK) mission is to unite the UK's health data to enable discoveries that improve people's lives.[1] One aspect of this activity is the ambition to provide a consistent view on the utility

### STRENGTHS AND LIMITATIONS OF THIS STUDY

⇒ We are not aware of any other publication reviewing open and open-source data profiling tools using this level of rigour.
⇒ A range of freely available data profiling tools are capability mapped regarding utility for profiling health data sets.
⇒ Use of such data profiling software tools can help improve data quality by understanding the technical dimensions of a given health data set.
⇒ There may be other potentially suitable tools in existence that were not discovered and evaluated.
⇒ It was not always possible to find out information on individual tools from available documentation.

of particular datasets for specific purposes through an Innovation Gateway.[2] This would allow users to understand whether a dataset is likely to meet their needs, ahead of requesting access. One important aspect of the utility of a dataset relates to the technical dimensions of data quality,[3] as the consistent use of data quality metrics can facilitate comparison between datasets and, in addition, can demonstrate areas of potential improvement for data custodians. Data quality is frequently cited as a challenge in undertaking health research, as well as for other uses of health data.[4] Commonly used data quality dimensions in health include completeness, consistency, uniqueness, validity, accuracy and timeliness.[5]

There are a variety of approaches used for establishing the quality of health data, hindering wider use of data due to challenges in understanding and communicating the usefulness of the data.[6] In addition to domain-specific subject matter expertise, semiautomated analysis of datasets using data quality profiling software tools can assist the process, supporting increased awareness of

BMJ

data quality of datasets, completeness and consistency of data submissions, improved reliability, accuracy and auditability and ultimately 'better' more usable data over time. Data profiling is the process of reviewing source data, understanding the structure, content and interrelationships of elements, examining records to discover errors/issues relating to content and format, and understanding data distributions and other factors.[7] It is seen as an important step towards improving the quality and usefulness of data.[8] There are many challenges in profiling data, depending on the structure and format of the underlying data.[9]

Many software tools are available, with varied applicability and data profiling capability for healthcare data. The aims of this study were to identify and evaluate functionality and usability of existing openly available (either open source or free-to-use) data quality assessment tools for potential users across the health data research community with specific focus on data profiling capabilities. There are many studies looking at the effectiveness of tools for data analysis, but few that focus on data profiling or curation.[10] This research often focuses on libraries or packages available to users of a specific coding language.[11 12] Through this research we wanted to provide resources available to understand the data itself.

Technical data quality metrics across the dimensions described above represents only a subset of overall characteristics to describe usefulness, or utility, of a dataset. Other factors, such as source, provenance, time period, geographical coverage, etc, may determine the utility for a particular project, independent of any technical data quality metrics.[13] Furthermore, data in a given data set may have an acceptable level of quality for some contexts or use cases, for example, a student technical project, but the same data may be inadequate in other contexts, such as use for healthcare regulatory purposes, based on a range of factors. The concept of overall evaluation of dataset utility for specific use cases is becoming more widely recognised.[14]

## METHODS
### Study design
In order to evaluate existing freely available data profiling tools for potential use with health datasets, a desk-based activity was performed. This first required the identification of as many tools as possible that would be available without cost, followed by an initial evaluation of the identified tools against a range of broad criteria based on publicly available information regarding the tool functionalities. Following this evaluation, tools which scored highly in the areas of most interest for profiling of health datasets were tested on a synthetic health dataset to evaluate their capability in an objective way.

### Identification of tools
An initial scoping exercise was conducted to identify data profiling tools that were freely available. This included tools that were open-source and those that were proprietary but freely available (or having a functional freely available version). The tools were identified through web searches, with search terms of 'data processing tools', 'data quality tools', 'data profiling tools' and 'data curation tools' and inclusion criteria being the absence license restrictions, cost, lack of expert level user requirements and appropriateness of functionality as relates to health data quality. This was supplemented by discussion with individuals currently working in the sector and involved in data profiling and curation. This process resulted in 28 potential tools for initial evaluation, some of which were generic tools.

### Initial evaluation
In order to evaluate the tools, a general comparison matrix was developed based on criteria used previously for evaluating data quality tools.[15] EM identified individual functions drawing from Gartner and Data Management Association (DAMA) criteria, as well as suggesting further functions, which could be categorised into functional areas and major categories. EM and TH developed an initial categorisation of functional areas and major categories, and this was refined in collaboration with BG, SV and NS. The scoring matrix was developed as a feature tree, comprising 5 major categories and 14 minor functional areas, and a maximum score allocated for each area. The 28 tools were initially compared and categorised against the matrix using information from the available product documentation and data sheets (table 1).

Each tool was ranked based on key capabilities required to address the profiling aspects of data quality using the feature tree and scoring. Tools were assigned the available weighted scoring based on the ability to provide the function described, according to the information available. Each feature was scored using a binary system, either 0 or 5. An exception to this rule is the 'Connectivity to N data sources' where this feature is scored 3, 4 and 5 when a tool has connectivity to <3, <6 and >5 data sources, respectively. Scores for each of the five major category areas were converted to a percentage of the total available score for that area.

### In-depth evaluation
Following the initial evaluation, eight tools scored were selected for further, in-depth evaluation based on the data profiling major category score and functions (the focus of this process was to evaluate data profiling capabilities; other potential functionalities were recorded for interest as above but not used for ranking). The selected tools included: Knime, DataCleaner, Orange, WEKA, Pandas-profiling (Python), Aggregate Profiler, Talend Open Studio for Data Quality, WhiteRabbit. (RapidMiner and DQ analyser were excluded since they were limited free versions of paid-for tools. Since two python tools, Pandas Profiling and Anaconda, scored highly for profiling, only Pandas profiling was further evaluated since it is explicitly intended for data profiling. Finally, WhiteRabbit, Talend

**Table 1** Detailed scoring criteria per feature

| Feature tree | | | | | | Score |
|---|---|---|---|---|---|---|
| Data ingestion and integration | → | Data consolidation | → | | Connectivity to N data sources | 5 |
| | | | → | | Data extraction, transformation and loading (ETL) and ETL support | 5 |
| | | | → | | Data modelling | 5 |
| | → | Data propagation | → | | Data flow orchestration, enterprise application integration, exchange of messages and transactions | 5 |
| | | | → | | Enterprise data replication, transfer large amounts of data between databases | 5 |
| | | | → | | Versioning and file management | 5 |
| | → | Data virtualisation | → | | Data access | 5 |
| | → | Data federation | → | | Enterprise information integration | 5 |
| | | | | | **Total** | | **40** |
| Data preparation and cleaning | → | Parsing and standardisation | → | | Tagging data with keywords, descriptions or categories | 5 |
| | | | → | | (Data scrubbing)/cleansing/handling blank values/reformatting values/ threshold checking | 5 |
| | | | → | | Data enhancement/enrichment/curation | 5 |
| | | | → | | Natural language processing | 5 |
| | | | → | | Address validation/geocoding | 5 |
| | | | → | | Master data management | 5 |
| | | | → | | Data masking | 5 |
| | → | Identity resolution, linkage, merging and consolidation | → | | Data deduping | 5 |
| | | | → | | Machine learning/training a statistical model | 5 |
| | | | → | | Data aggregation | 5 |
| | | | → | | Data binning | 5 |
| | | | → | | Grouping similar data/clustering | 5 |
| | | | → | | Outlier detection and removal | 5 |
| | → | Master reference data management | → | | 'Hub' infrastructure to source and distribute master/reference data | 5 |
| | | | → | | Master data versioning based on data history and timelines | 5 |
| | | | → | | Workflow integrations to steward and publish the master/reference data | 5 |
| | | | → | | Graph data stores to define relationships for creating a flexible knowledge graph | 5 |
| | | | → | | Accessible API (Application Programming Interface) for real-time access to shared reference data | 5 |
| | | | | | **Total** | | **90** |

**Table 1** Continued

| Feature tree | | | | | Score |
|---|---|---|---|---|---|
| Data profiling, exploration/ pattern detection | → | Relationship discovery | → | Cross-table redundancy analysis | 5 |
| | | | → | Performing data quality assessment, risk of performing joins on the data | 5 |
| | | | → | Identifying distributions, key candidates, foreign-key candidates, functional dependencies, embedded value dependencies and performing intertable analysis. | 5 |
| | → | Content discovery | → | Data pattern discovery | 5 |
| | | | → | Domain analysis | 5 |
| | | | → | Discovering metadata and assessing its accuracy | 5 |
| | → | Structure discovery | → | Column value frequency analysis and statistics, collecting descriptive statistics like min, max, count and sum. | 5 |
| | | | → | Table structure analysis, collecting data types, length and recurring patterns. | 5 |
| | | | → | Drill-through analysis | 5 |
| | | | | **Total** | **45** |
| Data monitoring | → | Monitoring and alerting | → | Time series data identified and collection by metric name and key/value pairs | 5 |
| | | | → | Flexible query language to leverage this dimensionality | 5 |
| | | | → | Graphing and dashboarding support | 5 |
| | | | | **Total** | **15** |
| Data use | → | Metadata management | → | Concept identification and naming | 5 |
| | | | → | Data categorisation | 5 |
| | | | → | Lineage | 5 |
| | | | → | Relationship with other metadata | 5 |
| | | | → | Comments and remarks | 5 |
| | | | → | Data statistics (profiles) | 5 |
| | | | → | Knowledge graph | 5 |
| | → | Privacy and security | → | Data anonymisation | 5 |
| | | | → | Role based access control | 5 |
| | | | → | Secure environment setup and deployment | 5 |
| | | | → | Container-based deployment | 5 |
| | → | Data mining | → | Interactive data visualisation | 5 |
| | | | → | Visual programming and analysis | 5 |
| | | | → | Visual illustrations and training documentation | 5 |
| | | | → | Sample data/generate fake data | 5 |
| | | | → | Add-ons and extension functionality | 5 |

Open Studio for Data Quality and Aggregate Profiler were also evaluated since they were identified as being used by the HDR UK community). To evaluate these tools for their data profiling performance and capability, synthetic data sets were created using the open source tool, Synthea to generate CSV files and SQL Database adhering to the Observational Medical Outcomes Partnership Common Data Model (an internationally adopted data standard) containing 1000 patients and related clinical data and the tools run on this dataset.[16] Synthea allows generation of

fully synthetic datasets which broadly conform to the data types and values expected in a 'real' health dataset but with no risk of patient data identification.[17] To evaluate performance and scalability of each tool an additional synthetic dataset of 1.3 million records was also generated.

Each of the shortlisted open-source data profiling tools were evaluated based on how possible it was to execute common specific profiling functions as described in the tool documentation decided based on the Gartner reports.[18]

Further to the initial evaluation, the shortlisted tools were evaluated in-depth based on the ability to deliver data profiles against core DAMA UK data quality dimensions,[3] including completeness (the proportion of stored data against the potential of 100% complete), consistency (the absence of difference, when comparing two or more representations of a thing against a definition), uniqueness (nothing recorded more than once based on how that thing is identified), validity (data are valid if it conforms to the syntax (format, type, range) of its definition), accuracy (the degree to which data correctly describes the object or event being described) and timeliness (the degree to which data represent reality from the required point in time). For each data profiling functionality, tools were run and subjectively scored on a scale of 0–5 according to a semistructured scale (0=unable to process, 1=most requirements not achieved, 2=some requirements not achieved, 3=meets core requirements, 4=meets and exceeds some requirements, 5=significantly exceeds core requirements).

The suitability of the tools for potential future use by other parties was estimated based on feedback from volunteers from the HDR UK community testing selected tools on their local datasets and providing a qualitative comment on usability. Formal evaluation of the tools of a range of real-world health datasets in a range of environments was outside the scope of this study.

## Patient and public involvement

Patients and/or the public were not involved in the design, or conduct, or reporting, or dissemination plans of this research.

## RESULTS
### Initial evaluation

The initial 28 tools evaluated are shown in online supplemental material 1 along with scores in the various data quality task categories with detailed results for data profiling functionality. The overall results of the initial scoring are shown in figure 1, where scores have been normalised to a maximum of 1 to support initial inspection.

### Subsequent evaluation

Based on the in-depth review of the selected eight tools to evaluate their ability to deliver key functions, the Python library, Pandas Profiling, was identified as possessing the most versatile functionality, able to complete all 30 of the identified profiling functions on the synthetic dataset for testing. The next most versatile tool, Knime, was able to perform 19 such tasks. Across the functionality types, Single Column—Cardinalities was one that the most tools were capable of delivering, with all tools able to deliver

| | Data Ingestion and Integration | Data Ingestion and Integration | Data Ingestion and Integration | Data Ingestion and Integration | Data Preparation and Cleaning | Data Preparation and Cleaning | Data Preparation and Cleaning | Data Preparation and Cleaning | Data Preparation and Cleaning | Data Profiling, Exploration / Pattern Detection | Data Monitoring | Data Use | Data Use | Data Use | Data Use |
|---|---|---|---|---|---|---|---|---|---|---|---|---|---|---|---|
| | Connectivity | Parsing | Issue resolution and workflow | Architecture and integration | Master Reference Data Management | Standardization and cleansing | Matching, linking and merging | Address validation / geocoding | Data curation and enrichment | Data profiling, measurement and visualization | Monitoring | Metadata management | Usability | DevOps environment | Deployment environment |
| Knime | 0.29 | 1.00 | 1.00 | 0.75 | 0.00 | 1.00 | 1.00 | 1.00 | 1.00 | 1.00 | 1.00 | 0.43 | 0.67 | 0.00 | 0.00 |
| Pandas Profiling | 0.29 | 0.00 | 0.00 | 0.00 | 0.00 | 0.00 | 0.00 | 0.00 | 0.00 | 1.00 | 0.00 | 0.00 | 0.33 | 0.00 | 0.00 |
| Orange | 0.29 | 1.00 | 1.00 | 0.25 | 0.00 | 0.50 | 1.00 | 1.00 | 0.67 | 1.00 | 1.00 | 0.00 | 0.67 | 0.00 | 0.00 |
| RapidMiner | 0.29 | 1.00 | 0.50 | 0.50 | 0.00 | 0.50 | 1.00 | 0.00 | 0.33 | 1.00 | 1.00 | 0.00 | 0.67 | 0.00 | 0.00 |
| WEKA | 0.18 | 0.00 | 0.00 | 0.00 | 0.00 | 0.25 | 0.80 | 0.00 | 0.67 | 1.00 | 0.00 | 0.43 | 0.17 | 0.00 | 0.00 |
| Anonimatron | 0.29 | 0.00 | 0.00 | 0.00 | 0.00 | 0.00 | 0.00 | 0.00 | 0.00 | 0.00 | 0.00 | 0.00 | 0.17 | 0.00 | 0.00 |
| ARX Data Anonymization | 0.29 | 0.00 | 0.00 | 0.00 | 0.00 | 0.25 | 0.00 | 0.00 | 0.33 | 0.00 | 0.00 | 0.00 | 0.33 | 0.00 | 0.00 |
| WhiteRabbit | 0.59 | 0.00 | 0.50 | 0.25 | 0.00 | 0.00 | 0.00 | 0.00 | 0.00 | 0.11 | 0.33 | 0.00 | 0.33 | 0.00 | 0.00 |
| Aggregate Profiler (AP) | 0.29 | 0.00 | 0.00 | 0.00 | 0.00 | 0.00 | 0.60 | 1.00 | 0.67 | 0.78 | 1.00 | 0.43 | 0.17 | 0.00 | 0.00 |
| Talend Open Studio for Data Integration | 0.29 | 1.00 | 0.50 | 0.25 | 0.00 | 0.00 | 0.00 | 0.00 | 0.00 | 0.00 | 0.00 | 0.00 | 0.00 | 0.00 | 0.00 |
| Talend Open Studio For Big Data | 0.29 | 1.00 | 0.50 | 0.25 | 0.00 | 0.00 | 0.00 | 0.00 | 0.00 | 0.00 | 0.00 | 0.00 | 0.00 | 0.00 | 0.00 |
| Talend Open Studio For Data Quality | 0.29 | 1.00 | 0.00 | 0.00 | 0.00 | 0.25 | 0.40 | 0.00 | 0.67 | 0.56 | 0.00 | 0.00 | 0.00 | 0.00 | 0.00 |
| Talend Open Studio For ESB | 0.29 | 0.00 | 1.00 | 0.00 | 0.00 | 0.00 | 0.00 | 0.00 | 0.00 | 0.00 | 0.00 | 0.00 | 0.00 | 0.00 | 1.00 |
| Talend Open Studio For MDM | 0.29 | 0.00 | 0.00 | 0.00 | 0.40 | 0.25 | 0.00 | 0.00 | 0.33 | 0.00 | 0.00 | 0.00 | 0.17 | 0.00 | 0.00 |
| OpenRefine | 0.18 | 1.00 | 0.00 | 0.25 | 0.00 | 0.25 | 0.80 | 0.00 | 0.00 | 0.00 | 0.00 | 0.00 | 0.00 | 0.00 | 0.00 |
| DataCleaner | 0.29 | 1.00 | 1.00 | 0.50 | 0.00 | 1.00 | 1.00 | 1.00 | 1.00 | 1.00 | 1.00 | 0.00 | 0.33 | 0.00 | 0.00 |
| DataPreparator | 0.18 | 0.00 | 0.00 | 0.25 | 0.00 | 0.25 | 0.60 | 0.00 | 0.00 | 0.00 | 0.00 | 0.00 | 0.00 | 0.00 | 0.00 |
| Data Match | 0.29 | 0.00 | 0.00 | 0.00 | 0.00 | 0.00 | 0.40 | 0.00 | 0.00 | 0.00 | 0.00 | 0.00 | 0.00 | 0.00 | 0.00 |
| DataMartist | 0.29 | 1.00 | 0.00 | 0.00 | 0.00 | 0.25 | 0.20 | 0.00 | 0.00 | 0.11 | 0.00 | 0.00 | 0.17 | 0.00 | 0.00 |
| Pentaho Kettle | 0.29 | 1.00 | 0.00 | 0.00 | 0.00 | 0.00 | 0.00 | 0.00 | 0.00 | 0.00 | 0.00 | 0.00 | 0.00 | 0.00 | 0.00 |
| SQL Power Architect | 0.29 | 0.00 | 0.00 | 0.25 | 0.00 | 0.00 | 0.00 | 0.00 | 0.00 | 0.00 | 0.00 | 0.14 | 0.00 | 0.00 | 0.00 |
| SQL Power DQguru | 0.29 | 0.00 | 0.00 | 0.00 | 0.00 | 0.50 | 0.60 | 1.00 | 0.00 | 0.00 | 0.00 | 0.00 | 0.00 | 0.00 | 0.00 |
| DQ Analyzer | 0.29 | 0.00 | 0.00 | 0.00 | 0.00 | 0.00 | 0.00 | 0.00 | 0.00 | 0.00 | 1.00 | 0.00 | 0.00 | 0.00 | 0.00 |
| Pimcore | 0.00 | 0.00 | 0.00 | 0.00 | 1.00 | 0.25 | 0.00 | 0.00 | 0.00 | 0.00 | 0.00 | 0.00 | 0.00 | 0.00 | 0.00 |
| CytoScape | 0.00 | 0.00 | 0.00 | 0.25 | 0.00 | 0.00 | 0.00 | 0.00 | 0.00 | 0.00 | 0.00 | 0.14 | 0.50 | 0.00 | 0.00 |
| Anaconda | 0.29 | 0.00 | 0.00 | 0.25 | 0.00 | 0.00 | 0.00 | 0.00 | 0.33 | 1.00 | 1.00 | 0.00 | 0.50 | 0.00 | 0.00 |
| pyxplorer | 0.24 | 0.00 | 0.00 | 0.00 | 0.00 | 0.00 | 0.00 | 0.00 | 0.00 | 0.11 | 0.00 | 0.00 | 0.00 | 0.00 | 0.00 |
| MobyDQ | 0.29 | 0.00 | 0.50 | 0.00 | 0.00 | 0.00 | 0.00 | 0.00 | 0.00 | 0.11 | 0.67 | 0.00 | 0.00 | 0.00 | 0.00 |

**Figure 1** Main results of documentation based functionality for data quality categories by tool.

three of the functions in this type. The functionality type that was least well served by the tools was Dependencies, with only Pandas Profiling able to deliver any of these functions (table 2).

The tools were further evaluated based on their ability to deliver data profiles against the DAMA dimensions (figure 2). Pandas Profiling achieved significantly greater results compared with the other tools, scoring 110 of the available points, compared with the next highest tool, Knime, with 61 points. Of the tools examined, WhiteRabbit had the least comprehensive functionality in this area, able only to provide information against the Completeness element. Across the different elements, completeness was best served by the profiling tools, with all tools able to provide some functionality in this area. The least well-served element was Consistency, with only Pandas Profiling able to provide any output for this element. Online supplemental material 2 shows the profile reporting information produced by Pandas Profiling with features including basic dataset statistics overview, reports on specific numerical or categorical variables and correlations between variables.

Links for all tools tested are available here (https://github.com/HDRUK/data-utility-tools).

### User testing feedback

To provide anecdotal feedback on the usability of the tools, five of the eight tools (DataCleaner, Orange, MobyDQ, Knime and Aggregate profiler) were tested by volunteers from the Cystic Fibrosis Trust and the Neonatal Medicine Research Group. These tools were selected for testing based of the volunteer's ability and the resources available to run them.

MobyDQ and Aggregate Profiler both presented difficulties to the volunteers due to challenges installing and running the software. MobyDQ failed to authenticate due to issues with private keys and Aggregate Profiler crashed on attempts to update.

Knime, DataCleaner and Orange could be run successfully by the volunteers. Orange required the local migration of data and installation of two additional modules, and was supported more effectively on Mac OS and Linux than Windows. Knime was fairly resource intensive and initially difficult to use, but was seen to be capable of a range of functions. DataCleaner was reported to be relatively easy to set up and run, even on a Windows machine, and capable of linking to existing databases.

### DISCUSSION

The findings of this study have demonstrated that numerous openly available data profiling tools are available, with several able to perform well using health datasets. The precise choice of tool for organisations will depend on the data type, model and format, in addition to Information Technology environment, such as Windows or Linux, and expertise with such tools and coding languages, such as Python. Regardless of the tools used, appropriate deployment and dataset evaluation through data profiling should lead to early detection of data quality issues for particular data sets and sources and consequent ability to remediate such issues. The identification of Pandas Profiling as a versatile approach to data profiling is reinforced by the fact that, as a Python library, it can be combined with other tools, such as Orange or Knime, to provide an even more in-depth output.

This study provides a useful resource for individuals anywhere in the world to understand the functionality of freely available data profiling tools for use with health datasets, and put these to use. The creation of an open and persistent resource is a strength of the study. All the outputs of the testing, as well as the generated dataset, are available (https://github.com/HDRUK/data-utility-tools). None of the tested tools are specific to health data, and therefore could be used in any other domain. However, the open nature of the search for the tools, the absence of an indexed repository of these tools was likely non-exhaustive. There may be additional tools that would also have been suitable for this exercise that were not identified during the project. Furthermore, the tools were tested on a synthetic dataset, which was useful for testing functionality, but does not necessarily represent the condition of 'real' health data, which may include numerous additional or unexpected errors and anomalies. Ideally, the team would have been able to test the tools on real patient data, but information governance approvals were not possible in the available time and a fully standardised dataset was required to ensure objectivity when comparing tools, hence a controlled synthetic dataset was most appropriate for the present purposes. While some of the tools were tested on real datasets by volunteers (Cystic Fibrosis Trust and Neonatal Data Analysis Unit), this was designed to review the initial views regarding usability of the tool, rather than provide a comparison of the outputs.

Determining data quality is a complex process and far harder than commonly assumed, especially for high dimensional and longitudinal data such as health data. Data profiling provides the user with an understanding of the inherent technical data quality according to various dimensions within a given dataset but does not, in itself, improve quality. Rather, based on the outcome of data profiling, it will likely be required to use one or more data quality tools to remediate issues detected, this being best accomplished by data analysts and/or scientists with subject matter expertise, working close to the original source of the data. While the ability of the tools to be used by individuals with limited experience was not the focus of this research, this would be interesting to explore in future work, particularly because the tool with the broadest capability, Pandas Profiling, was not tested by volunteers. There are a large number of libraries and packages available for coding languages such as Python and R, for example, skimr.[19] These resources provide powerful capabilities for analysts, but often require some

**Table 2** Specific data profiling tool functionalities evaluated

| Functionality type | Function | Data profiling tools capable of natively executing function | | | | | | | |
| --- | --- | --- | --- | --- | --- | --- | --- | --- | --- |
| | | K | DC | O | W | PP | AP | TOS | WR |
| **Single Column-Cardinalities** Refers to the uniqueness of data values contained in a particular column (Attribute) of a table (Entity) | No of rows | ✓ | ✓ | ✓ | ✓ | ✓ | ✓ | ✓ | ✓ |
| | No of nulls | ✓ | ✓ | ✓ | ✓ | ✓ | ✓ | ✓ | ✓ |
| | Percentage of nulls | ✓ | | ✓ | ✓ | ✓ | | ✓ | ✓ |
| | No of distinct values (cardinality) | ✓ | ✓ | ✓ | ✓ | ✓ | ✓ | ✓ | ✓ |
| | Percentage of distinct values (No of distinct values divided by the no of rows) | ✓ | | | ✓ | ✓ | | ✓ | |
| **Single Column-Value distribution** Presents an ordering of the relative frequency (count and percentage) of the assignment of distinct values | Frequency histograms (equi-width, equi-depth, etc.) | ✓ | | | | ✓ | | | |
| | Minimum and maximum values in a numeric column | ✓ | ✓ | ✓ | | ✓ | ✓ | ✓ | ✓ |
| | Constancy (Frequency of most frequent value divided by number of rows) | ✓ | | | | ✓ | | ✓ | |
| | Quartiles (three points that divide the numeric values into four equal groups) | ✓ | ✓ | | | ✓ | ✓ | ✓ | ✓ |
| | Distribution of first digit in numeric values (to check Benford's law) | ✓ | | | | ✓ | | ✓ | |
| **Single Column-Patterns, datatypes and domains** Refers to the discovery of patterns and data types | Basic types (eg, numeric, alphanumeric, date, time) | ✓ | | | | ✓ | | | |
| | Database Management Systems-specific data type (eg, varchar, timestamp) | ✓ | ✓ | | | ✓ | ✓ | ✓ | ✓ |
| | Measurement of Value length (minimum, maximum, average, median) | ✓ | ✓ | ✓ | | ✓ | ✓ | | ✓ |
| | Maximum number of digits in numeric values | ✓ | ✓ | | | ✓ | ✓ | | |
| | Maximum number of decimals in numeric values | ✓ | | | | ✓ | ✓ | | |
| | Histogram of value patterns (Aa9…) | ✓ | ✓ | | | ✓ | | ✓ | |
| | Generic semantic data type (eg, code, date/time, quantity, identifier) | ✓ | ✓ | | | ✓ | | ✓ | |
| | Semantic domain (eg, credit card, first name, city) | ✓ | ✓ | | | ✓ | | ✓ | |
| **Dependencies** Determines the dependent relationships within a data set | Unique column combinations (key discovery) | | | | | ✓ | | | |
| | Relaxed unique column combinations | | | | | ✓ | | | |
| | Inclusion dependencies (foreign key discovery) | | | | | ✓ | | | |
| | Relaxed inclusion dependencies | | | | | ✓ | | | |
| | Functional dependencies | | | | | ✓ | | | |
| | Conditional functional dependencies | | | | | ✓ | | | |
| **Advanced Multi Column profiling** Determines the similarities and differences in syntax and data types between tables (entities) to determine which data might be redundant and which could be mapped together | Correlation analysis | | | ✓ | | ✓ | ✓ | | |
| | Association rule mining | | | | | ✓ | | | |
| | Cluster analysis | | | | | ✓ | | | |
| | Outlier detection | ✓ | | ✓ | | ✓ | | | |
| | Exact duplicate tuple detection | | ✓ | | | ✓ | | ✓ | |
| | Relaxed duplicate tuple detection | | ✓ | | | ✓ | | ✓ | |
| | Total | 19 | 13 | 8 | 5 | 30 | 10 | 15 | 8 |

AP, Aggregate Profiler; DC, DataCleaner; K, Knime; O, orange; PP, Pandas Profiling (Python); TOS, Talend Open Studio for Data Quality; W, WEKA; WR, WhiteRabbit.

| 0 = Unable to process | 3 = Good: meets requirements |
| 1 = Poor: most or all defined requirements not achieved | 4 = Excellent: meets or exceeds some requirements |
| 2 = Fair: some requirements not achieved | 5 = Outstanding: significantly exceeds core requirements |

| Measure (key elements) | White Rabbit | Orange | Knime | WEKA | Aggregate Profiler | Data Cleaner | Pandas (Python) | Talend Open Studio - Data Quality |
|---|---|---|---|---|---|---|---|---|
| **COMPLETENESS - The proportion of stored data against the potential of "100% complete"** | | | | | | | | |
| Percentage of requisite information available | 2 | 4 | 4 | 3 | 2 | 3 | 5 | 1 |
| Percent of missing data values (null / empty string) | 2 | 4 | 4 | 4 | 3 | 3 | 5 | 1 |
| Row counts | 4 | 5 | 4 | 4 | 4 | 3 | 5 | 2 |
| Highest and lowest value of key elements | 0 | 3 | 5 | 0 | 0 | 3 | 5 | 1 |
| Number of data values in an unusable state | 0 | 2 | 2 | 0 | 0 | 3 | 5 | 0 |
| **UNIQUENESS - No thing will be recorded more than once based upon how that thing is identified.** | | | | | | | | |
| (Number of things in the real world) - Number of incorrect spellings etc. of same data in an element e.g. address (duplicate values) | 0 | 2 | 2 | 0 | 1 | 2 | 5 | 2 |
| (Number of recodes describing different things) Number of data items in adherence to expected/described data element value (distinct values at ID level) | 0 | 1 | 2 | 0 | 1 | 2 | 5 | 1 |
| (Number of things in real world i.e. duplicates)/(Number of records describing different things i.e. distinct records) | 0 | 3 | 4 | 4 | 1 | 2 | 5 | 1 |
| **TIMELINESS - The degree to which data represent reality from the required point in time.** | | | | | | | | |
| Difference between Lowest date value and Highest Date Value | 0 | 2 | 4 | 0 | 1 | 2 | 3 | 1 |
| Number of records per month | 0 | 1 | 3 | 0 | 0 | 2 | 3 | 0 |
| **VALIDITY - Data are valid if it conforms to the syntax (format, type, range) of its definition.** | | | | | | | | |
| Percentage of data values that comply with the specified formats (data types, ranges etc.) | 0 | 1 | 3 | 0 | 0 | 4 | 5 | 2 |
| Percentage of data values that don't comply to specified formats | 0 | 0 | 1 | 0 | 0 | 1 | 4 | 0 |
| Number of Missing values indicated e.g. with fill values | 0 | 4 | 4 | 0 | 4 | 3 | 5 | 2 |
| Number of Values in Specified Range | 0 | 0 | 3 | 0 | 0 | 3 | 4 | 0 |
| Number of values not in Specified Range | 0 | 0 | 2 | 0 | 0 | 3 | 3 | 0 |
| **ACCURACY - The degree to which data correctly describes the "real world" object or event being described.** | | | | | | | | |
| Number of accurate data values | 0 | 3 | 3 | 0 | 2 | 0 | 5 | 2 |
| Number of inaccurate data values | 0 | 0 | 0 | 0 | 0 | 0 | 5 | 0 |
| Actual data value count versus predicted data value count | 0 | 0 | 0 | 0 | 0 | 0 | 3 | 0 |
| Number of rows and columns against expectations | 0 | 0 | 0 | 0 | 0 | 0 | 3 | 0 |
| Number of duplicates at ID level | 0 | 4 | 4 | 4 | 3 | 3 | 5 | 3 |
| Number of blank columns, large % of blank data, high % of same data | 0 | 3 | 4 | 0 | 2 | 0 | 5 | 2 |
| Distribution across various segments | 0 | 3 | 0 | 0 | 0 | 0 | 5 | 0 |
| Outliers on key variables | 0 | 3 | 2 | 0 | 0 | 0 | 4 | 0 |
| ((Count of accurate objects)/ (Count of accurate objects + Counts of inaccurate objects) | 0 | 1 | 1 | 0 | 0 | 0 | 3 | 0 |
| **CONSISTENCY - The absence of difference, when comparing two or more representations of a thing against a definition.** | | | | | | | | |
| Analysis of pattern and/or value frequency | 0 | 0 | 0 | 0 | 0 | 0 | 5 | 0 |
| **TOTAL SCORES** | 8 | 49 | 61 | 19 | 24 | 42 | 110 | 21 |

**Figure 2** Results of profiling tasks using synthetic datasets. Knime and Pandas performed best for overall data profiling tasks for this healthcare dataset.

amount of technical capability, reducing their accessibility to many users.

Further research would be useful to understand the capability of the tools in handling increasingly large sets of data. While the tools were tested against a dataset of over one million patient records, processing time was not compared quantitatively. Further, in a healthcare or health research setting, it is not unusual for a dataset to be several orders of magnitude larger than this. For a tool to be useful in these settings, it should be able to process large datasets, and within a reasonable time.

As referenced in the Introduction, there is a need for greater consistency in how dimensions of data quality are assessed and communicated. The wider adoption of data profiling tools would encourage greater literacy and higher expectations among users of health data. Transparency of current dataset profiles, for example, on the Innovation Gateway, would provide an incentive for focused improvement of data, as well as informed decision making by users. Further work could be done in the presentation of the outputs of data profiling exercises, in order to ascertain the approach that is most conducive to effective data curation.

Evaluation of a wide range of freely available software tools for data engineering with a focus on data profiling for healthcare data tested using synthetic datasets has determined that several tools perform highly in a range of tasks appropriate to this use case. By the more widespread use of routine health dataset profiling, and associated remediation, along with other measures to understand and improve dataset utility, we anticipate that the overall quality of health data for research use can be increased.

**Contributors** BG, SV and NS conceived the study. EM, TH, OD, RJ and VR developed the methodology further, evaluated the tools and provided the initial results. KE and VB tested the tools on their own datasets and provided feedback on results. NS, BG, CF and JB prepared and drafted the manuscript. The guarantor of the content is NS.

**Funding** This work was supported by Medical Research Council capital funding (August 2019). There is no grant number associated with capital fund awards.

**Competing interests** EM, TH, OD, RJ and VR were employed by Inspirata at the time of the work but were contracted by HDR UK to carry out this work independently on behalf of HDR UK.

**Patient and public involvement** Patients and/or the public were not involved in the design, or conduct, or reporting, or dissemination plans of this research.

**Patient consent for publication** Not applicable.

**Provenance and peer review** Not commissioned; externally peer reviewed.

**Data availability statement** Data are available on reasonable request. Links for all tools tested, as well as documentation from the review are available here: https://github.com/HDRUK/data-utility-tools.

**ORCID iDs**
Ben Gordon http://orcid.org/0000-0002-3105-6774
Susheel Varma http://orcid.org/0000-0003-1687-2754
Jake Barrett http://orcid.org/0000-0003-4758-6400

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
