## [Reviewer comments · BMJ Open]

ARTICLE DETAILS

TITLE (PROVISIONAL)	Evaluation of Freely Available Data Profiling Tools for Health Data Research Application: a functional evaluation review
AUTHORS	Gordon, Ben; Fennessy, Clara; Varma, Susheel; Barrett, Jake; McCondochie, Enez; Heritage, Trevor; Duroe, Oenone; Jeffery, Richard; Rajamani, Vishnu; Earlam, Kieran; Banda, Victor; Sebire, Neil

VERSION 1 – REVIEW

REVIEWER	Opalek, Amy Foundation for Advancement of International Medical Education and Research
REVIEW RETURNED	25-Aug-2021

GENERAL COMMENTS	This manuscript, Evaluation of Freely Available Data Profiling Tools for Health Data Research Application, presents a detailed, comparative analysis of the characteristics of several tools used for the automated evaluation of data sets in the specific context of health data. The paper is clearly written, and the purpose is well defined. The tables and results demonstrate that the authors achieved their goals. On its face, this research provides practical, actionable guidance to readers on the selection of a data profiling tool. However, this work is not grounded in current research on data quality in medical/clinical informatics and may not be accessible to BMJ Open's readers in its current form. Introduction: The authors provide appropriate references to data management literature, but these sources and the field in general may not be germane to many BMJ Open readers. There is a wealth of research on health data quality specifically, and this should be addressed in the introduction. The value of the authors' research depends upon demonstrating its necessity in the health data realm, supported by the literature. Methods: This lack of engagement with health/medical/clinical informatics literature may also be problematic from a methodological standpoint. A review of this specific literature could reveal additional tools developed specifically in the health data space which might have been worthy of review (e.g., https://doi.org/10.1016/j.ijmedinf.2018.02.007). It is not clear whether the web searches described in the "Identification of Tools" section included a review of the clinical informatics literature. There may also be data quality evaluation frameworks within clinical informatics that include criteria not found in Gartner and DAMA frameworks used, and the choice of framework should be justified or at least acknowledged as a discretionary choice. Please spell out and reference the OMOP data model for clarity.
--

	Results: The data tables, supplementary material, and analysis of results are very informative and well presented. The “User Testing” portion of the study, though not the focus, provides some extra insight that is warranted. (After all, if the software is impossible to install or run, its extensive functionality is moot.) Discussion: The discussion section of this manuscript aptly summarizes the findings. However, points and references are raised here that should have been part of the introduction (paragraphs 4 and 5), and there is again the issue of lack of clinical informatics engagement. In particular, this statement needs to be supported in the introduction: “[u]ptake of routine profiling of data is not yet commonplace within the health data sector...” Overall, the authors have shown considerable, thoughtful effort in analyzing the capabilities of many available data profiling tools in the context of a health data set. This could be an important contribution to the field, but the authors need to demonstrate this more clearly for the BMJ Open audience.
--	--

REVIEWER	Noertjojo, Kukuh WorkSafe BC, Clinical Services
REVIEW RETURNED	20-Dec-2021

GENERAL COMMENTS	I appreciate very much for the changes you made to the previous suggestions for this paper that I think is quite practical and relevant for data science study. I recommend for your paper to be published but I have a question to you:  - how do you develop the scoring matrix (Table 1); is there any reason for the scoring? Did you validate and test for its reliability prior to deploying it in this paper. This matrix is very important part of your paper and I think it is important to provide the evidence on the utility of this tool.. - it seems that KNIME, although the software is free but there is fee associated with its server use (https://www.knime.com/knime-software/knime-server-pricing). You may need to take this into account/explain it in your paper. - I did not see any statement on Conflict of Interest from the authors. This is perhaps relevant since your paper is assessing products albeit a "free" ones. Thank you and it's been a pleasure reading your paper.
---

REVIEWER	Le Meur, Nolwenn Ecole des Hautes Etudes en Sante Publique
REVIEW RETURNED	03-Jan-2022

GENERAL COMMENTS	In their paper “Evaluation of Freely Available Data Profiling Tools for Health Data Research Application” the authors propose an interesting scoring system to assess “quality” of data profiling tools that could be used for healthcare data. Although the method is well described, the results are difficult to related with the method and lack of clarity. Major comments ----- Globally for the different tables and figures, it is difficult to see what
---

	scoring system (if any) is used. I do not see the relationship between the criteria proposed in the methods section and the elements of results. This need to be clarified. For instance, it remains unclear whether the scoring system is binary (0 or 5) or on a scale from 0 to 5. Page 9 and 10 the Authors wrote that: Each feature was scored using a binary system, either 0 or 5. An exception to this rule is the “Connectivity to N data sources” where this feature is scored 3, 4, and 5 when a tool has connectivity to < 3, < 6, and > 5 data sources, respectively. But page 11 the Authors say: “For each data profiling functionality, tools were run and subjectively scored on a scale of 0-5 according to a semi-structured scale (0=unable to process, 1=most requirements not achieved, 2=some requirements not achieved, 3=meets core requirements, 4=meets and exceeds some requirements, 5=significantly exceeds core requirements). “ P44 is it Figure 1? what is the scale of the scoring??? How does it correlate with what is proposed in the method section? Figure 1 is insufficiently explained. Figure 2. The 0 is labelled as “not applicable” but if it is an expected/wanted functionality would not it be more appropriate to say “unavailable” or “missing functionality”? It is somehow a shame that Panda was not tested by naïve users as it seems to be the best performing tools. A last criterion on the level of expertise in computer science, bioinformatic or global IT environment might be interesting to add as installation all tools are not equivalent. In addition, as mentioned by the authors, awareness should be raised on the need for data scientist or statistician expertise around data profiling phase and a good knowledge on the origin and goals of the dataset in hands. In addition, no word is said on processing time and scalability. The Authors mentioned it in the method section (p11), but no results are shown. Could the authors elaborate on those criteria that might of interest as we handle more and more massive health data. Minor comments ----- In the Introduction line 5, what is HRD? If it corresponds to Health Data Research, it should be written in full letter the first time. In table 1, does NLP stand for natural language processing? Why sometimes lower letters and sometimes capital letters in pair of terms? Ex Data Deduping or Data Binning but Data aggregation.
--	--

VERSION 1 – AUTHOR RESPONSE

Reviewer: 1

Introduction: The authors provide appropriate references to data management literature, but these sources and the field in general may not be germane to many BMJ Open readers. There is a wealth of research on health data quality specifically, and this should be addressed in the introduction. The value of the authors’ research depends upon demonstrating its necessity in the health data realm,

supported by the literature.

The introduction has been re-written to more clearly situate the question within the context of health data, referencing relevant sources. To clarify, this article is not a review regarding health data quality and general health data management but is rather a specific and targeted evaluation of openly available data profiling tools for health data research, and the scope is limited to this.

Methods: This lack of engagement with health/medical/clinical informatics literature may also be problematic from a methodological standpoint. A review of this specific literature could reveal additional tools developed specifically in the health data space which might have been worthy of review (e.g.,

<https://gbr01.safelinks.protection.outlook.com/?url=https%3A%2F%2Fdoi.org%2F10.1016%2Fj.ijmedinf.2018.02.007&data=04%7C01%7Cben.gordon%40hdruk.ac.uk%7C4d1aa9597b7a44214aed08d9d83590c1%7C844cacb1702742639d8a18faa5bf0839%7C1%7C0%7C637778546838232017%7CUnknown%7CTWFPbGZsb3d8eyJWIjoiMC4wLjAwMDAiLCJQIjoiV2luMzliLCJBTiI6Ikl1haWwiLCJXVCi6Mn0%3D%7C3000&sdata=x1vih4EG7BjCjYKJcw3pXsliUgQRK1J4XFv96WY8Q1pk%3D&reserved=0>). It is not clear whether the web searches described in the “Identification of Tools” section included a review of the clinical informatics literature. There may also be data quality evaluation frameworks within clinical informatics that include criteria not found in Gartner and DAMA frameworks used, and the choice of framework should be justified or at least acknowledged as a discretionary choice. Please spell out and reference the OMOP data model for clarity.

More detail has been given on the choice of tools. The rationale for the choice of approach and framework chosen has been included in the Methods section, and the OMOP common data model referenced. Please note, as in the point above this manuscript is targeted on openly available general purpose data profiling tools, it is not a systematic review of all literature related to research data management. For example, the reference suggested by the reviewer re ICU-DAMA is not a widely available general purpose data profiling tool and hence has not been included. We now try to further clarify this scope in the manuscript.

Discussion: The discussion section of this manuscript aptly summarizes the findings. However, points and references are raised here that should have been part of the introduction (paragraphs 4 and 5), and there is again the issue of lack of clinical informatics engagement. In particular, this statement needs to be supported in the introduction: “[u]ptake of routine profiling of data is not yet commonplace within the health data sector...”

Thank you for this suggestion. Detail from the paragraphs in question have been moved to the introduction, and the sentence altered to more directly reflect the context

Reviewer: 2

How do you develop the scoring matrix (Table 1); is there any reason for the scoring? Did you validate and test for its reliability prior to deploying it in this paper. This matrix is very important part of your paper and I think it is important to provide the evidence on the utility of this tool.

Thank you. The scoring table was developed specifically for this exercise, to provide a pragmatic structure for testing the tools. More detail regarding scoring has now been provided on the approach in the Methods section. We are not suggesting this scoring matrix be used more widely, rather it was used to provide structure of comparing tools.

- it seems that KNIME, although the software is free but there is fee associated with its server use (<https://gbr01.safelinks.protection.outlook.com/?url=https%3A%2F%2Fwww.knime.com%2Fknime-software%2Fknime-server-pricing&data=04%7C01%7Cben.gordon%40hdruk.ac.uk%7C4d1aa9597b7a44214aed08d9d83590c1%7C844cacb1702742639d8a18faa5bf0839%7C1%7C0%7C637778546838232017%7CUnknown%7CTWFPbGZsb3d8eyJWIjoiMC4wLjAwMDAiLCJQIjoiV2luMzliLCJBTiI6Ikl1haWwiLCJXVCi6Mn0%3D%7C3000&sdata=FHFk9KUCxmctLAAZo9D5PkwIJ6ABau10ySGHb7j6G%2FI%3D&reserved=0>). You may need to take this into account/explain it in your paper.

There is a charge associated with KNIME Server, however there is a free, open-source version of the tool (KNIME Analytics), which is what was evaluated as part of this project.

- I did not see any statement on Conflict of Interest from the authors. This is perhaps relevant since your paper is assessing products albeit a "free" ones.

Thank you. We have added a conflicts of interest section at the end of the paper

Reviewer: 3

In their paper "Evaluation of Freely Available Data Profiling Tools for Health Data Research Application" the authors propose an interesting scoring system to assess "quality" of data profiling tools that could be used for healthcare data.

The aim of the manuscript was to compare functional capabilities of freely available data profiling tools for healthcare data sets. We do not propose a scoring system to assess quality of data profiling to be used outside of this study, merely to provide a pragmatic framework for the study itself. We have now clarified this in the manuscript

It remains unclear whether the scoring system is binary (0 or 5) or on a scale from 0 to 5. Page 9 and 10 the Authors wrote that: Each feature was scored using a binary system, either 0 or 5. An exception to this rule is the "Connectivity to N data sources" where this feature is scored 3, 4, and 5 when a tool has connectivity to < 3, < 6, and > 5 data sources, respectively. But page 11 the Authors say: "For each data profiling functionality, tools were run and subjectively scored on a scale of 0-5 according to a semi-structured scale (0=unable to process, 1=most requirements not achieved, 2=some requirements not achieved, 3=meets core requirements, 4=meets and exceeds some requirements, 5=significantly exceeds core requirements). "

Further clarification has been added to the paper to outline that the scoring system on pages 9 & 10 (and results in Figure 1) refers to the initial evaluation of all tools, whereas the scoring approach on page 11 refers to the in-depth reviewing of the shortlisted tools

P44 is it Figure 1? what is the scale of the scoring??? How does it correlate with what is proposed in the method section? Figure 1 is insufficiently explained.

More detail has now been given in the Methods and Results section on the approach taken, and further explanation has been given on Figure 1 in the text

Figure 2. The 0 is labelled as "not applicable" but if it is an expected/wanted functionality would not it be more appropriate to say "unavailable" or "missing functionality"?

Figure 2 has been updated to reflect that 0 means "unable to process", as indicated in the text

It is somehow a shame that Panda was not tested by naïve users as it seems to be the best performing tool. A last criterion on the level of expertise in computer science, bioinformatic or global IT environment might be interesting to add as installation all tools are not equivalent. In addition, as mentioned by the authors, awareness should be raised on the need for data scientist or statistician expertise around data profiling phase and a good knowledge on the origin and goals of the dataset in hands.

This is a helpful point and has been included in the discussion

In addition, no word is said on processing time and scalability. The Authors mentioned it in the method section (p11), but no results are shown. Could the authors elaborate on those criteria that might of interest as we handle more and more massive health data.

This function was not explored quantitatively as part of this study, and therefore no further information can be provided however this limitation has been referenced in the Discussion

In the Introduction line 5, what is HRD? If it corresponds to Health Data Research, it should be written in full letter the first time.

Thank you, this has now been expanded

In table 1, does NLP stand for natural language processing?

Yes, this has now been expanded

Why sometimes lower letters and sometimes capital letters in pair of terms? Ex Data Deduping or Data Binning but Data aggregation.

All terms have now been expressed in sentence case

VERSION 2 – REVIEW

REVIEWER	Noertjojo, Kukuh WorkSafe BC, Clinical Services
REVIEW RETURNED	22-Feb-2022

GENERAL COMMENTS	I thank you for the revision that you have done based on the reviewers suggestion. This study represent a practical and useful tools for health data reserachers
--

REVIEWER	Le Meur, Nolwenn Ecole des Hautes Etudes en Sante Publique
REVIEW RETURNED	01-Mar-2022

GENERAL COMMENTS	The paper has been improved in terms of readability. It is now clear that the aim of the paper is the comparison of functional capabilities of freely available data profiling tools for healthcare data sets. The scoring matrix (although interesting for further research) is only a tool for them to perform their comparison. Although questions remain on the searching process and inclusion/selection criteria for the tools to compare. (1) What keywords were used for the web searches? “Data profiling” might not be the only term used for (first) data quality assessment. It can also be “data (pre-)processing”, “data quality assessment”, “data exploration” and maybe more. Could the authors list their keywords, even in supplementary materials. (2) As a corollary, I am surprised that no R packages has been tested as a previous work by Staniak, Mateusz and P. Biecek. (“The Landscape of R Packages for Automated Exploratory Data Analysis.” R J. 11 (2019): 347.) explored a similar topic. It is true that R is a command line software but as is Panda and it is not more complex to install than MobyDQ. In addition, R is commonly used in healthcare research. Finally, as mentioned by the authors, this review does not pretend to be exhaustive on the tools that might be available but citing similar works (as the one indicated above) is necessary. The bibliographical section is centered on the functional capabilities to be investigated but lack of references on similar benchmarking studies on the same tools or similar ones.
--

VERSION 2 – AUTHOR RESPONSE

Reviewer: 2

Dr. Kukuh Noertjojo, WorkSafe BC

Comments to the Author:

I thank you for the revision that you have done based on the reviewers suggestion. This study represent a practical and useful tools for health data reserachers

Thank you for your response

Reviewer: 3

Dr. Nolwenn Le Meur, Ecole des Hautes Etudes en Sante Publique Comments to the Author:

The paper has been improved in terms of readability. It is now clear that the aim of the paper is the comparison of functional capabilities of freely available data profiling tools for healthcare data sets. The scoring matrix (although interesting for further research) is only a tool for them to perform their comparison.

Although questions remain on the searching process and inclusion/selection criteria for the tools to compare.

(1) What keywords were used for the web searches? "Data profiling" might not be the only term used for (first) data quality assessment. It can also be "data (pre-)processing", "data quality assessment", "data exploration" and maybe more. Could the authors list their keywords, even in supplementary materials.

Additional information has been included in the Methods section to identify the search terms used.

(2) As a corollary, I am surprised that no R packages has been tested as a previous work by Staniak, Mateusz and P. Biecek. ("The Landscape of R Packages for Automated Exploratory Data Analysis." R J. 11 (2019): 347.) explored a similar topic. It is true that R is a command line software but as is Panda and it is not more complex to install than MobyDQ. In addition, R is commonly used in healthcare research.

The paper suggested is a useful addition, and has been included in the introductory section (see response to comment below). As the reviewer mentions, R is a powerful tool and useful to many analysts. Reference has been made to R packages, such as skimr, in the paper, with a further emphasis of the non-exhaustive nature of the research, and the likelihood that if a user is able to utilise tools of this nature, they are unlikely to require the packages explored in this research.

Finally, as mentioned by the authors, this review does not pretend to be exhaustive on the tools that might be available but citing similar works (as the one indicated above) is necessary. The bibliographical section is centered on the functional capabilities to be investigated but lack of references on similar benchmarking studies on the same tools or similar ones.

This is a helpful suggestion, and the introductory section has been developed to highlight other work of this nature, including the paper referenced above and similar publication from 2018 looking at free python libraries for data analysis, noting that in both of these cases the purpose of the research was the use of the software for data analysis, rather than meta-analysis and assessment of data ahead of undertaking specific analysis.